# Leak State Detection and Size Identification for Fluid Pipelines with a Novel Acoustic Emission Intensity Index and Random Forest

**DOI:** 10.3390/s23229087

**Published:** 2023-11-10

**Authors:** Tuan-Khai Nguyen, Zahoor Ahmad, Jong-Myon Kim

**Affiliations:** Department of Electrical, Electronics, and Computer Engineering, University of Ulsan, Ulsan 44610, Republic of Korea; khaint@mail.ulsan.ac.kr (T.-K.N.); zahooruou@mail.ulsan.ac.kr (Z.A.)

**Keywords:** acoustic emission, Bayesian ensemble, b-value, changepoint detection, industrial pipeline, intensity index, leak detection, leak size identification, random forest

## Abstract

In this paper, an approach to perform leak state detection and size identification for industrial fluid pipelines with an acoustic emission (AE) activity intensity index curve (AIIC), using b-value and a random forest (RF), is proposed. Initially, the b-value was calculated from pre-processed AE data, which was then utilized to construct AIICs. The AIIC presents a robust description of AE intensity, especially for detecting the leaking state, even with the complication of the multi-source problem of AE events (AEEs), in which there are other sources, rather than just leaking, contributing to the AE activity. In addition, it shows the capability to not just discriminate between normal and leaking states, but also to distinguish different leak sizes. To calculate the probability of a state change from normal condition to leakage, a changepoint detection method, using a Bayesian ensemble, was utilized. After the leak is detected, size identification is performed by feeding the AIIC to the RF. The experimental results were compared with two cutting-edge methods under different scenarios with various pressure levels and leak sizes, and the proposed method outperformed both the earlier algorithms in terms of accuracy.

## 1. Introduction

Pipelines play a vital role in industrial settings as one of the major fluid transportation methods. While they can generally provide safe and efficient passage even in long-distance systems, pipelines are not immune to leaking and breaking due to the influence of natural disasters, component corrosion, installation errors, etc. An unattended leak can evolve into a major pipeline burst, which can not only sever the entire transportation system, but potentially lead to human injuries/fatalities, environmental hazards, and huge economic costs for recovery. Due to the dire consequences of pipeline leaks, it is essential that the faults are detected and diagnosed in earlier stages, so that maintenance can be performed accordingly.

Currently, non-destructive testing (NDT) has emerged as one of the most popular research topics within the prognosis and health management framework for civil and industrial assets [1,2,3]. In addition to its obvious contribution to safety and incident prevention, the other reason behind NDT’s relevance is that it allows inspection and evaluation without affecting specimen serviceability, as its name suggests. Regarding NDT in pipeline systems, multiple techniques have been proposed such as ultrasonic testing (UT) [4,5,6], radiography testing (RT) [7,8], ground-penetrating radar (GPR) [9], infrared thermography testing (IFT) [10], distributed acoustic sensing [11], active distributed temperature sensing [12], acoustic emission (AE) [13,14,15,16,17,18], etc. Each of these techniques presents various advantages and disadvantages which can best serve different applications. In terms of AE, it studies the elastic waves released upon a discontinuity occurrence. AE offers many upsides, such as non-directionality, no downtime for in-service testing, new damage detection, and outstanding progression monitoring, in which the deterioration process can be captured with only one test. During the monitoring process, AE is sensitive to new damage sustained by the specimen, and to the irregularities of the specimen’s working activity due to these damages. Since the foremost purpose of this study is pipeline monitoring, it can benefit from the aforementioned advantages. Thus, AE was the NDT method chosen for application in this work.

### 1.1. Related Works

Many studies have been conducted that have used AE for pipeline system diagnosis [19,20,21,22,23,24,25,26,27,28,29,30,31,32,33]. When a leak occurs in a pipeline system, variations are expected in the AE data. The leak detection process using AE is centered around the differences between the variations that happen in the leak state and the normal state, which have been mostly approached from the view of feature extraction and pattern recognition in recent studies [20,21,25,26,27,28,29,30,31,32,33,34,35,36,37,38,39]. The studies in this area can be roughly categorized as follows: time domain [25,26,34,35], frequency domain [36,40] and time–frequency domain [27,32,37,38,39].

Time domain approaches are often the most straight forward out of the three domains; however, they usually require an additional pre-processing step to minimize their susceptibility to interferences. Noseda et al. [25] proposed an approach to transition time estimation, in combination with the power of a neural network, to devise a propagation classification. AE features, such as AE counts, cumulative AE energy, etc. were investigated and utilized, along with support vector machine (SVM) and relevance vector machine (RVM) in [26], for the identification and localization of a pipeline leak. The authors in [34] presented a solution, using artificial neural networks (ANN) with amplitude data from time domain AEs. As for the study in [35], the authors investigated the leak detection problem by using statistical time domain features, which were selected by principal component analysis and then fed to a support vector data description model for working state classification.

Frequency domain approaches can be more robust than the time domain ones; however, they are more complicated and often require more expertise regarding system- or fault-specific frequencies. Through the analysis of leak-related frequencies’ amplitude, the authors in [40] proposed a solution for leak detection and leak size identification. In addition, the study in [36] presented an approach for leak detection by harnessing AEs’ spectral portraits to obtain encoded spectral envelope features. Because frequency domain analysis is more favorable towards stationary signals, difficulties might arise due to the gas/fluid pipeline AE signal being non-stationary [31,41].

Out of the three domains, time–frequency approaches can provide the most powerful solutions, at the cost of huge computational complexity. The most notable methods in this domain are wavelet transform [32,37], empirical mode decomposition [27], and variational mode decomposition [39], etc. The features, once extracted by the previously mentioned methods, are then harnessed by different artificial intelligence techniques for state diagnosis, such as neural network [42], SVM [26,37,38], or adversarial network [28], etc.

As briefly explained in the previous subsection, AE is based on the release of elastic waves upon the occurrence of discontinuity. Such an occurrence, which releases elastic waves, is called an AE event (AEE) and, when an AEE is detected by a sensor, these data can be referred to as an AE hit (AEH). Generally, AEHs can be categorized into two types: burst-type and continuous-type [15]. While the continuous-type is considered noise and contains minimal useful information, the burst-type AEH possesses valuable data that can be used for the determination of the specimen’s state. Because of the nature of fluid transportation in pipeline systems, AE events can be recorded from various sources, even under normal working conditions that can involve unstable flow, pressure on joints, pipeline vibration, leaks, and interferences from the environment, etc. The diversity of sources in pipeline systems and their occurrences in a short span of time introduce a more complex problem (referred to as the multi-source problem of AEEs), in comparison to rigid specimens such as concrete structures and machinery, etc. in which the signal is significantly more transient and there are fewer sources of AEEs (primarily due to discontinuity-related reasons, such as cracks, loose fittings, etc.). Even though AEH features can offer a great description of the AE activity, it is difficult to correctly obtain the data related to these features due to the high AE noise level and the sophisticated AEH detection process. As an alternative, this paper presents AE activity intensity construction, using the b-value for pipeline systems. The b-value was first introduced in the Gutenberg–Richter law (GRL) [43]. In brief, it expresses the relationship between the number of earthquakes and their magnitude in a given time period and location. Since an AEE can be considered a low-energy seismic event [44], the b-value was also adopted for AE applications on concrete structures [45,46,47]. As the adopted b-value describes the correlation between the number of AE peaks and the AE amplitude, and there is a proven relationship between the amplitude and the discontinuity size [48,49], it is anticipated to provide an accurate description for the AE activity intensity. Based on this intensity index, the pipeline’s working condition can be detected and, if a leak is present, the size can be identified. A more detailed walkthrough of the proposed method is presented in the next subsection.

### 1.2. Overview of the Proposed Method and Contributions

The proposed method’s simplified flowchart is presented in Figure 1. The raw AE data were recorded from the testbed using three AE sensors. Following the collection of raw AE data, a pre-processing step was conducted with a bandpass filter to keep only the desired frequency range. Then, the filtered data were utilized to extract the b-value and construct the AE activity intensity index curves (AIICs). Afterward, the AIICs were harnessed for leak state identification (normal working condition or leakage) using a Bayesian ensemble estimator for changepoint detection, which calculated the probability of the occurrence of a changing event. Once the leakage was confirmed, the AIIC parts that corresponded to the leaking state with multiple leak sizes and pressure levels were separated and fed to a random forest (RF) model for leak size classification training/testing. The proposed method demonstrated high accuracy under various conditions, even in comparison to cutting-edge references.

The summary of this work’s contributions is as follows:The AIIC was constructed using the b-value, which provides a robust and descriptive representation of the industrial pipeline’s working conditions, addressing the multi-source problem of AEE in pipeline systems.Pipeline leak state detection was implemented using the proposed AIIC, along with changepoint detection using a Bayesian ensemble estimator, and size identification using RF.The validation was performed using AE for industrial fluid pipelines under various leak sizes and pressure levels.

The organization of this paper is as follows:Section 1 (Introduction) presents the motivation for and background of this study, along with related works, an overview of the proposed method and the contributions.Section 2 (Basic Concepts) presents the background for the algorithms utilized in this study, including the b-value, changepoint detection using a Bayesian ensemble estimator, and RF.Section 3 (Case Study) provides a detailed explanation of the proposed method and its validation, along with the description of the experimental setup and dataset.Section 4 (Conclusion) concludes the research and discusses future work.

## 2. Basic Concepts

### 2.1. The b-Value

The GRL b-value is one of the most important parameters for the probabilistic investigation of seismic hazards. It is the slope of a log-normal distribution of the size of seismic events, or the relationship between the event magnitude and the frequency of occurrence. In a particular region within a specific period, the GRL can be defined as follows:(1)$$\log_{10}N=a-bM,$$
in which a is an empirical constant, b is the b-value, and N is the number of seismic events with a magnitude higher than M. Due to an AE being a high-frequency but low-energy seismic activity, the GRL is adaptable to AE analysis with a few modifications, as follows:(2)$$\log_{10}N(A)=a-b\frac{A}{20},$$
with a being an empirical constant (which is set at zero in this study), b being the b-value, and N(A) being the number of AE hits with an amplitude equal to or larger than the amplitude threshold A (dB). A higher b-value indicates that the number of AE peaks with a low amplitude is large, along with the low level of AE activity. These AE activities mostly come from pipeline vibration, unstable flow, or AE noise, etc., from which little useful information can be presumably obtained. In contrast, a lower b-value indicates the presence of multiple AE peaks with high amplitude and a more intense level of AE activity, which can be witnessed in the visualization of the relationship between event magnitude and frequency of occurrence provided in Figure 2. In theory, this can be associated with a discontinuity in the fluid pipeline, given that the AE peak amplitude is related to the size of the discontinuity and that the AEEs occurring due to a leak are releasing more energy than those under normal conditions [49]. Thus, it is expected that, upon initialization of a leak, a significant decrease in the b-value will be recorded and the b-value can help discern the difference between leak sizes.

The amplitude threshold governs the amount of information the b-value can convey. Given a high threshold with only a minimal number of higher AE peaks or a threshold lower than the majority of AE peaks, the b-value would not be able to capture the change in AE peak amplitude distribution over time. Therefore, it is of paramount importance to obtain a suitable value for the threshold. Variations in the AIICs, which are constructed from the b-value, can be expected in normal conditions due to multiple sources of AEEs and the hard threshold nature of the b-value.

### 2.2. Changepoint Detection

Trend analysis and changepoint detection can be considered complementary approaches that both target changes in data. Trend analysis detects abrupt changes and nonlinear dynamics in time series data [50,51,52] and is a dominant research topic, particularly in geoengineering. One of the most formidable challenges for time series analysis is the inconsistency in different model findings regarding the same problem during the decomposition of time series. This encouraged the idea of combining multiple models in the statistical literature, which can be traced back to Roberts’s proposal of a distribution of two models combined (1965). Later, a scheme was designed to evaluate the likelihood of each model in a collection being the best and then using that to average them into a single model, which was later known as Bayesian model averaging (BMA) [53]. In this study, the Bayesian Estimator of Abrupt change, Seasonal change, and Trend (BEAST) by Zhao et al. [51], which was originally developed to handle environmental time series data, was implemented for leak detection. The idea of this method is to exploit the relative value of decomposition models as individuals via averaging them using BMA. Given a time series ${\left\{t_{i},y_{i}\right\}}_{i=1,\ldots ,N}$, BEAST decomposes it into different components containing seasonality, trend, and changepoints, along with noise, as follows:(3)$$y=S\left(t_{i};\Theta_{S}\right)+T\left(t_{i};\Theta_{T}\right)+\varepsilon_{i},$$
in which $S\left(\cdot \right)$ and $T\left(\cdot \right)$ are seasonal and trend signals, respectively; ε is noise with a Gaussian distribution; and $\Theta_{s}$ and $\Theta_{T}$ have changepoints (a major changepoint is also referred to as a true abrupt change) implicitly encoded within them. General linear models were applied for $S\left(\cdot \right)$ and $T\left(\cdot \right)$ parameterization, with $S\left(\cdot \right)$ being a piecewise harmonic model and $T\left(\cdot \right)$ being a piecewise linear function.

As the pipeline working state changes from normal to leaking, not only will AEEs arise from the discontinuity occurrence on the pipeline surface itself, but flow turbulences are also expected. This further increases the number of high-amplitude AE activities and as such, changes the distribution of the frequency and the amplitude. With the shift of AE activity towards the region with a higher amplitude, a decrease in the b-value should be seen. Since the AIIC can show fluctuations in the pipeline’s normal working state as previously mentioned, any sudden drop in value might be misdiagnosed as a changepoint, which would mean a leak initialization transitioning the pipeline from normal conditions into the leaking state. In addition, the underlying processes which change the distribution in fluid pipeline AE signals are nonlinear, rather than being purely linear or piecewise-linear. For these reasons, BEAST can perform better than those constructed from either of the previously mentioned views.

### 2.3. Random Forest

Despite providing good interpretability at a shallow depth, decision trees (DTs) are susceptible to the overfitting problem and high variance as they deepen. This hinders DTs from giving a good performance for real-world problems, which often go along with huge datasets. Eventually, this shortcoming led to the birth of RF, which was first proposed by Breiman in 2001 [54], inspired by the work of Amit and Geman [55]. The idea of RF is simple: it exploits a DT efficiency with a smaller dataset by training each DT separately with a bootstrapped dataset, and then aggregates these DTs together for a conclusion [56,57]. The simplified classification process using RF is displayed in Figure 3.

Given a real-valued input vector $X={\left(X_{1},\ldots ,X_{p}\right)}^{T}$ and a real-valued response Y=f(X), the goal is to find the prediction function f(X), which is determined by a loss function $L(Y,f\left(X\right))$. For classification problems, zero-one loss is often used for L:(4)$$L\left(Y,f\left(X\right)\right)=\left\{\begin{array}{c} \;1\;if\;Y\neq f(X)\\ 0\;otherwise \end{array}\right.,$$

Then, the problem becomes finding the expected value of the loss $E_{XY}(L\left(Y,f\left(X\right)\right))$ minimization. Given $\bar{Y}$ as the set containing possible Y values, the minimization problem gives:(5)$$f\left(x\right)=\arg\max P(Y=y\;|\;X=x),$$
with $y\in \bar{Y}$. This is also known as the Bayes rule. Each of the DT returns a base learner ($h_{1}\left(x\right),\ldots h_{N}\left(x\right)$) and by majority vote, the ensemble predictor $f\left(x\right)$ can be predicted as:(6)$$f\left(x\right)=\arg\max\sum_{i=1}^{N}I(y=h_{i}\left(x\right)).\;$$

## 3. Case Study

### 3.1. Proposed Method

For the overview of the proposed method, the summary can be revisited in Section 1.2 and Figure 1. The process starts with the preprocessing of raw AE data, collected from the pipeline system. A bandpass filter from 10^5^ to 3.5 × 10^5^ Hz was designated for this purpose. This process is necessary to minimize the unwanted frequency bands and to keep only the frequencies in which beneficial leak-related information can be extracted.

Once the filtered data were obtained, b-values were calculated next, using Equation (2). For monitoring, this computation cyclically occurs over a one-second period, to construct the AIIC. As previously discussed, choosing the amplitude threshold for b-value calculation is very important. Analysis of the frequency–amplitude distribution (FAD) was conducted to find a suitable value. A visualization of the FAD is demonstrated in Figure 4, Figure 5 and Figure 6, for the normal state transitioning into the leaking state under the different pressure levels of 7 bar, 13 bar, and 18 bar, respectively. In each of these scenarios, the pipeline worked under the normal condition at the beginning until the 120th cycle, the point at which the leak started, and the pipeline then started working under the leaking condition. More details about the experimental setup with different scenarios are listed in Section 3.2. As can be seen across the pressure levels, the FADs were similar under the normal working condition, with the majority of AE activity happening in a low-amplitude regime, and only a rather small portion coming from AEEs with a higher amplitude (whose presence is understandable, given that there are sources of AEEs other than leaks). However, when a leak happened, the FAD shifted towards a higher-amplitude region, which can be easily spotted as the pressure levels and leak sizes increased. During this period, a noteworthy rise in the number of high-amplitude AE activities was witnessed, which was no mere coincidence as leak-related AE activities were the largest source of these. The differences in FAD for various leak sizes can also be noticed. For the case of low pressure and a micro leak size such as in Figure 4a, the rise is still noticeable from 10^−2^ and above, despite it being not as significant as the rest. The FAD analysis showed that the amplitude threshold value at 10^−2^ can best describe the AE activity under different scenarios.

Following AIIC construction, leakage detection was the next target, for which BEAST was implemented. By analyzing the trend, abrupt changes, and the nonlinear dynamics of the AIIC, it detected the changepoint of the working state (from normal working condition to leakage) by predicting the occurrence probability of a change in the AIIC. The prior probability was obtained following the formulas provided in [51]. A visualization of the detection of leak initialization using BEAST can be found in Figure 7.

If the pipeline is detected as being in the leaking stage, then the leak size identification process, using RF, begins. The optimal hyperparameters for RF were achieved by using grid search, along with cross-validation, and are displayed in Table 1. Finally, a majority voting strategy was utilized to obtain the final size identification result. This can provide a better generalization than a single DT, and therefore minimizes the risk of overfitting [58]. A dataset description for training/testing is presented in Section 3.2.

### 3.2. Experimental Setup and Dataset Description

To validate the approach presented in this paper, data acquisition was conducted using an industrial fluid pipeline system. Three R15I-AST sensors were deployed during this process. The R15I-AST sensor is a highly sensitive product from MITRAS, specifically designed to provide low-noise input, with a built-in auto sensor test and long-cable driving capability without an additional preamplifier requirement. Due to its metal housing, it is less susceptible to radio frequency and electromagnetic interferences. The three sensors were attached to the surface of the pipeline following application of a coupling medium gel on the contact spots. To ensure that the sensors would not be displaced or fall from the specimen, tape was utilized. Sensor channels one, two and three were located at 2500, 1600, and 0 mm (reference position), respectively, along with the leak location at 800 mm. The flow direction was from channel one to channel two, then through the leak position, and finally through channel three. The multiple sensor setup allowed different points of view: channel one offered the view of an upstream flow before the leak; channel two was used for near-leak monitoring, which was expected to have the highest AE activity intensity of all of the channels; channel three offered a view of a downstream flow located after the leak. Though they were recording from the same test, each of them could be treated as standalone, which provides more variety to the dataset. An image and schematic of this experimental setup is displayed in Figure 8.

Prior to the main acquisition, each sensor underwent a sensitivity test and calibration with the Hsu-Nielsen test [59], whose simplified illustration is displayed in Figure 9. The Hsu-Nielsen test, also known as the pencil break test, utilizes the breaking of pencil lead, which is positioned on the specimen surface at 30 degrees, to reproduce an AEE. Prior to AE data acquisition, it is necessary to calculate the attenuation characteristics following the ISO standard 18211:2016 [60], whose computation is as follows:(7)$${Attenuation}_{dB}=20\log_{10}\frac{V}{V_{ref}},\;$$
in which, the attenuation is in decibels (dB); V and $V_{ref}$ refer to the respective measured and reference potentials. It is advised to have an AE sensor surveillance zone within a distance corresponding to 25 dB [15] to avoid severe signal strength loss, which amounts up to 10.9 m for this experimental setup, with R15I-AST sensors and a pipeline system with an outer diameter of 114.3 mm.

The data collected by AE sensors were first processed by a 16-bit analog to digital converter integrated with the National Instruments’ NI-9223 module. In combination with software developed specifically for data acquisition by Ulsan Artificial Intelligence Laboratory, the whole process was monitored and controlled. A detailed description of the setup can be found in Table 2.

A total of nine tests were conducted for this study, including three levels of pressure and three sizes of leak. Each test was initialized under the normal working condition for a period and then switched to the leaking state for the rest of the test with the activation of a simulated leak, whose size and operation status were controlled by a fluid control valve welded on a hole in the pipeline surface. For detailed information, Table 3 displays the data descriptions regarding the nine tests conducted for this study. In combination with the three sensors deployed in each test, 27 data streams of the fluid pipeline working under the normal state and then transiting to the leaking state were collected and were used for leak detection. After activation of a leak in the pipeline system, unstable flow was witnessed for a short time. For this reason, only the parts of data streams from the 126th cycle until the end were utilized for leak size identification. A window size of 10 cycles with a nine-cycle overlap was used to divide the leak-identification dataset into segments, which were then harnessed to train and test RF. For training and testing, the dataset was split with a ratio of 70%/30%. During the training period, the training set was further divided for fivefold cross-validation.

### 3.3. Validation of the Proposed Method

In this subsection, validation of the proposed method is divided into three different validations: the AIIC constructed from b-value, leak detection using BEAST, and leak size identification using RF.

Firstly, the results of AIIC constructed from the b-value are displayed in Figure 10, Figure 11 and Figure 12. The AIIC does not just show a significant gap between normal and leaking conditions, but also the potential to discriminate between different leak sizes. As shown in Figure 4, Figure 5 and Figure 6, the FAD for various leak sizes differs, which then translates into the AIIC that was built based on the b-value. In the pipeline’s normal working state, fluctuations are witnessed due to the multiple sources with random contributions. However, these variations are witnessed less in the leaking state as the leak becomes the dominant source of all high-amplitude AE activities. With the high-amplitude leak-related AE activities greatly outnumbering other sources, the AIIC becomes less susceptible to the multi-source problem of the AEE. For further validation, a comparison was made between the AIIC and traditional statistical indexes, as shown in Figure 13. Among the traditional indexes, mean (Figure 13b) and kurtosis (Figure 13d) do not distinguish the difference between two working states; despite root mean Square (RMS, Figure 13c), standard deviation (STD, Figure 13e), and variance (Figure 13f) being able to show a certain level of distinction between normal condition and leakage, wild fluctuations can be witnessed in the later state, in comparison to the AIIC. This type of variation in the indexes, which is minimized using the AIIC, indicates that they are significantly affected by the high noise level, and therefore do not provide a great description of the pipeline’s working conditions.

For every scenario, a steep drop can be witnessed around the time when the leak would start, as expected and explained in Section 3.2. In this method, the cycle error using BEAST is calculated as follows:(8)$$cycle\;error=\left|t_{p}-t_{r}\right|,\;$$
with $t_{p}$ being the highest occurrence probability of the changepoint predicted by BEAST and $t_{p}>50\%$; and $t_{r}=120$ being the real leak activation time. Despite multiple significant decreases in AIIC prior to the leak, the changepoint can still be detected with only an average of a 2.6 cycle error and a standard deviation of 0.5 in all scenarios. This cycle error is impressive because it also consists of the time after leak activation when the fluid flow within the pipeline is unstable, which is often neglected, as the leaking stage is frequently considered to start upon stabilization of the flow.

Finally, leak size identification was conducted once the leakage was confirmed. A ratio of 70%/30% was used for the random train-test split from the dataset. Due to the AIIC’s capability for discrimination and the classification power of RF, it shows a high accuracy of approximately 100% for all levels of pressure. The results obtained from the classification process are shown in Figure 14.

For further validation, a comparison was conducted with two cutting-edge methods: (1) a method for size identification using convolutional neural networks (CNN) with time domain AE features (this reference method is referred to as TD-CNN) and (2) an adaptation of the method proposed by [26] using AEH-based features and a 1D-CNN (instead of their proposed SVM–RVM, which was originally developed for different pressure rates) to classify the leak sizes (this reference method is referred to as the Banjara et al. adaptation). All three methods, including the proposed one and the two references, were evaluated using the leak identification dataset. This leak identification dataset contained only the leaking state part from the 27 data streams (from the 126th cycle until the end of each stream, as explained in Section 3.2). The results for different pressure levels are shown in Table 4, Table 5 and Table 6.

The leak size identification results in Table 4, Table 5 and Table 6 show that the proposed method outperforms the two state-of-the-art references in terms of the average classification accuracy across different levels of pressure. Concerning the first comparison under a 7 bar pressure, TD-CNN shows a better average accuracy than the Banjara et al. adaptation (88.5% versus 77%); however, it is significantly lower than the 100% accuracy of the proposed method. The second and third comparisons demonstrate a difference, because the Banjara et al. adaptation excels in both scenarios (92.2% and 87.0%, respectively) against the TD-CNN (79.6% and 75.9%, respectively). Similar to the first case, the proposed method has the best result of the three methods. The reference methods show inconsistency with a classification accuracy up to 100.0% but also as low as 45.0% (Banjara et al.) or 64.4% (TD-CNN). This problem does not exist in the proposed method, which is anticipated because AIIC is more robust against the multi-source problem of AEEs than the other two, and efficiently discriminates leak sizes (Figure 10, Figure 11 and Figure 12). The time domain features in the TD-CNN and AEH features in the method of Banjara et al. (despite being capable of providing a great description of AE activity in more transient data) are more likely to be affected by the high level of background noise.

## 4. Conclusions

Pipelines play an essential role in fluid transportation, and pipeline leakage events can cause great damage to humans, the environment, and assets. Therefore, detection and diagnosis must be continually performed to ensure that maintenance can be planned and performed accordingly to prevent further loss. This study presents a leak state detection and leak size identification method for an industrial fluid pipeline, using the acoustic emission (AE) activity intensity index curve (AIIC) based on the b-value and random forest (RF). Due to AE being a high-frequency but low-energy seismic activity, the b-value, which is one of the most important parameters for the probabilistic investigation of seismic hazards in seismology, can be adapted to AE applications. Because the adapted b-value characterizes the relationship between the amplitudes and their frequencies of occurrence, along with there being a direct AE activities’ amplitude–intensity connection, an AIIC based on the b-value can parameterize the AE activity intensity in any time period. A frequency–amplitude distribution (FAD) analysis showed that the majority of high-amplitude AE activities in the leaking state were leak-related, which greatly outnumbered other sources that contributed to the background noise and multi-source problem of AE activities in a fluid pipeline system. The AIIC calculated from the b-value also discriminated between not only normal and leaking operation conditions, but also different leak sizes. In the next step, leak detection was performed using the AIIC through changepoint detection using a Bayesian ensemble, which showed an outstanding average error of only 2.6 s from the moment of leak occurrence. Once a leak was confirmed, the AIICs used for leak detection were again utilized for size identification using RF. The fluid pipeline data, with three pressures and three leak sizes, were used to validate the proposed method. The size identification results were compared with two cutting-edge methods, which showed that the proposed method can outperform the earlier algorithms with significantly higher accuracy. For future work, leak localization can be integrated into this study, along with different scenarios, such as pipelines buried under soil or submerged in water. In addition, the limitation of a hard-threshold b-value can also be further researched to achieve a better description of AE activity.

## Figures and Tables

**Figure 1 sensors-23-09087-f001:**
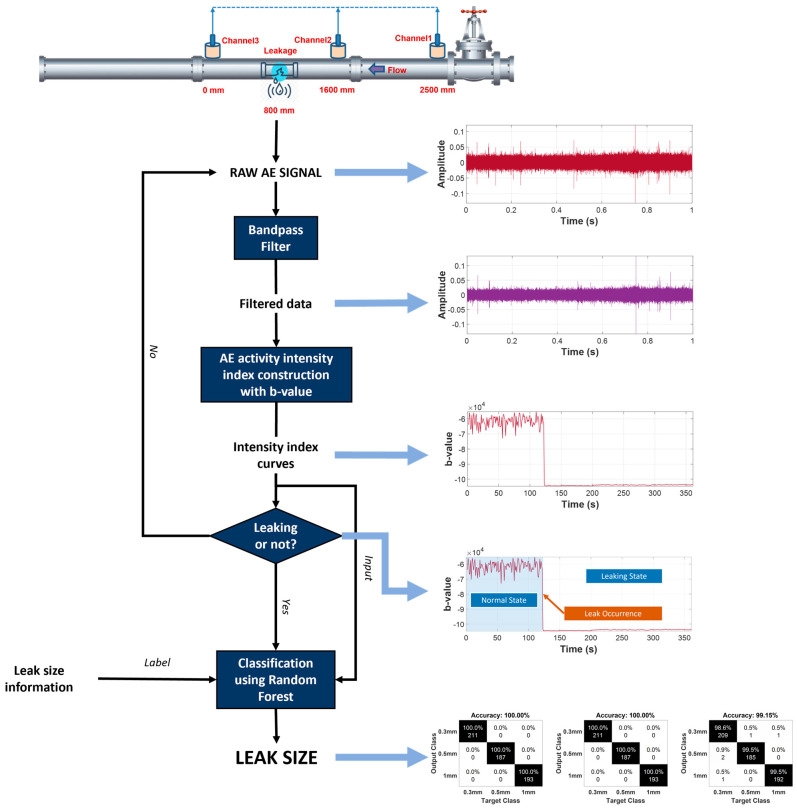
Flowchart of the proposed method.

**Figure 2 sensors-23-09087-f002:**
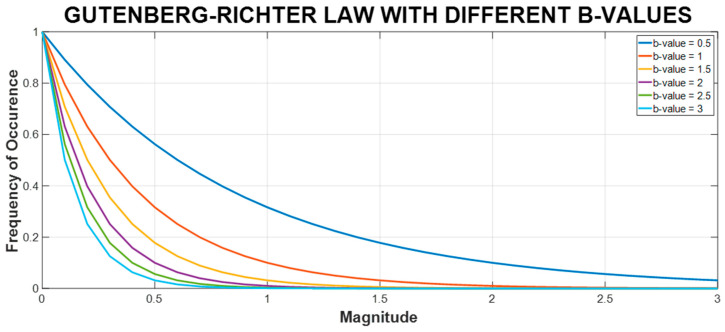
Visualization of magnitude vs. occurrence of different b-values based on GRL.

**Figure 3 sensors-23-09087-f003:**
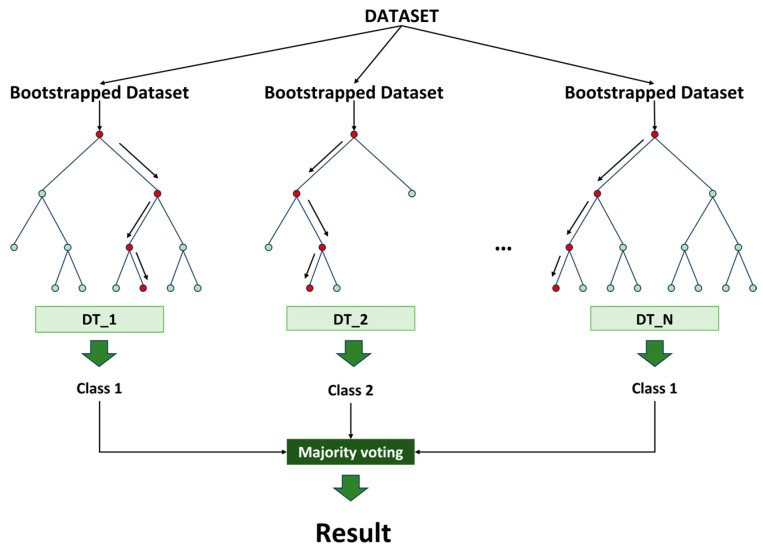
Simplified visualization of classification process in RF.

**Figure 4 sensors-23-09087-f004:**
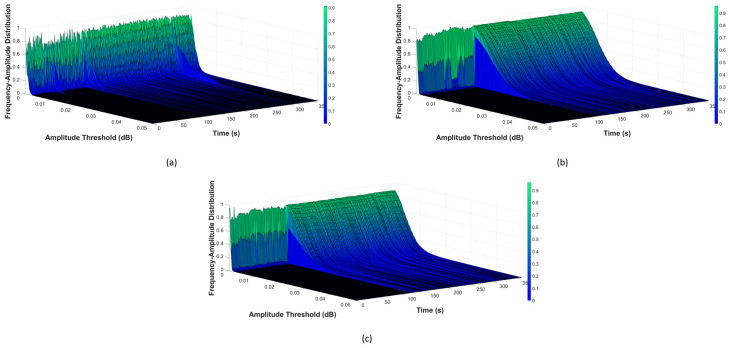
Visualization of FAD at pressure level 7 bar with different leak sizes: (**a**) 0.3 mm, (**b**) 0.5 mm, and (**c**) 1 mm.

**Figure 5 sensors-23-09087-f005:**
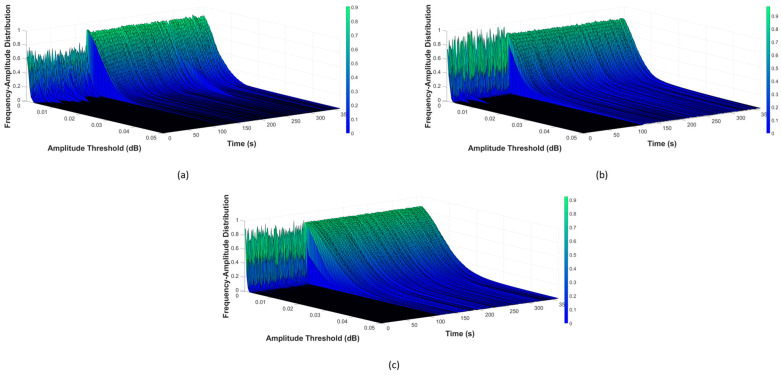
Visualization of FAD at pressure level 13 bar with different leak sizes: (**a**) 0.3 mm, (**b**) 0.5 mm, and (**c**) 1 mm.

**Figure 6 sensors-23-09087-f006:**
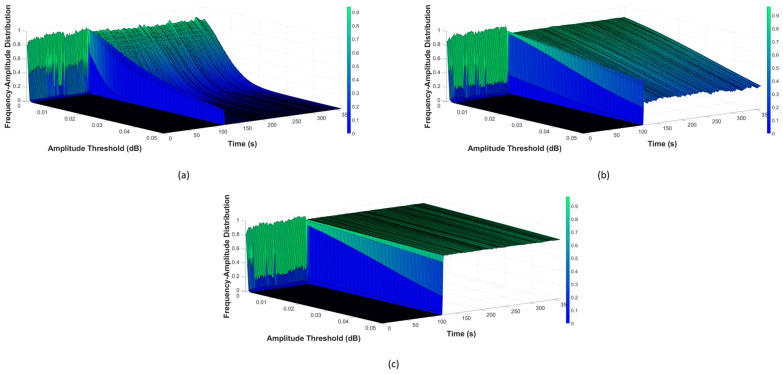
Visualization of FAD at pressure level 18 bar with different leak sizes: (**a**) 0.3 mm, (**b**) 0.5 mm, and (**c**) 1 mm.

**Figure 7 sensors-23-09087-f007:**
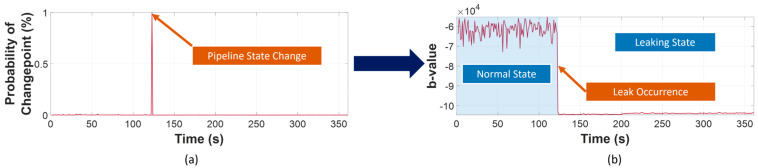
Leak detection using BEAST: (**a**) changepoint occurrence probability; (**b**) leak detected.

**Figure 8 sensors-23-09087-f008:**
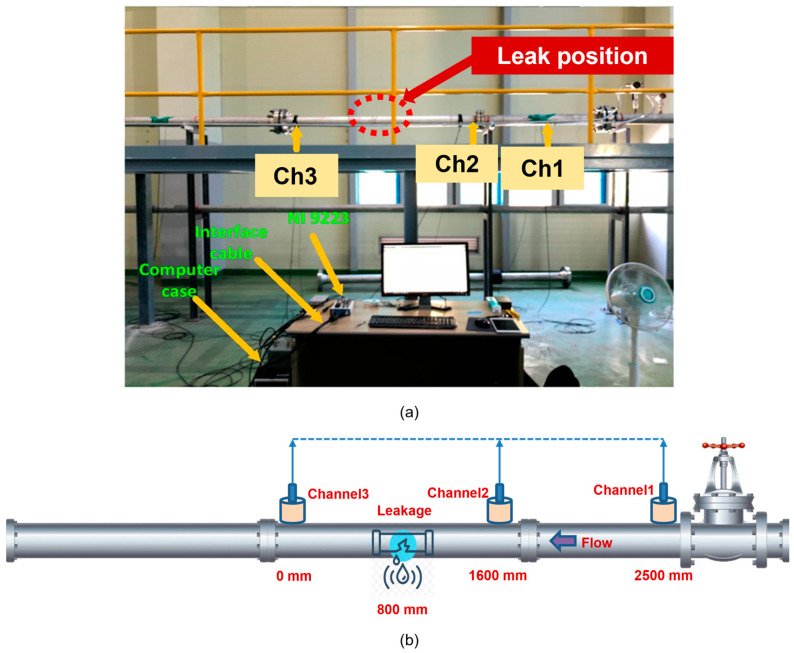
Experimental setup: (**a**) an image of the testbed and (**b**) an illustration of the sensor placements and leak position.

**Figure 9 sensors-23-09087-f009:**
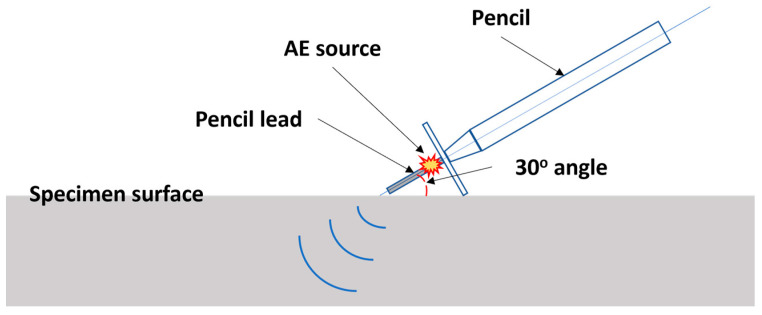
A simplified visualization of Hsu-Nielsen test.

**Figure 10 sensors-23-09087-f010:**
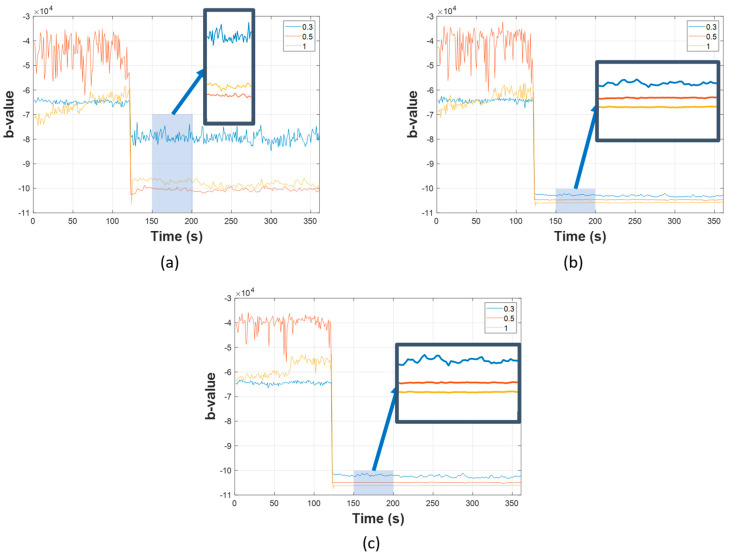
AIICs for different leak sizes at 7 bar for: (**a**) channel one, (**b**) channel two, and (**c**) channel three.

**Figure 11 sensors-23-09087-f011:**
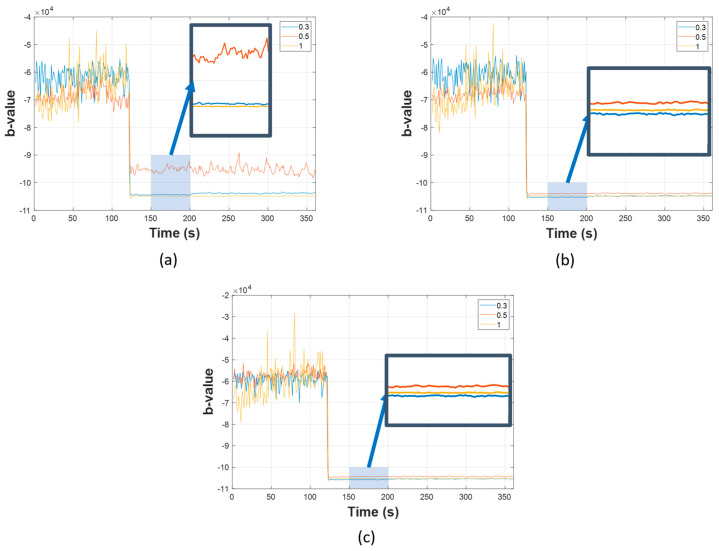
AIICs for different leak sizes at 13 bar for: (**a**) channel one, (**b**) channel two, and (**c**) channel three.

**Figure 12 sensors-23-09087-f012:**
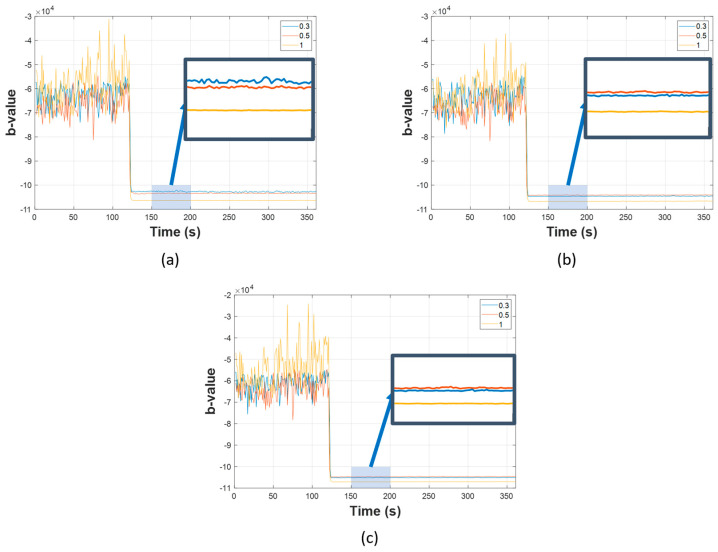
AIICs for different leak sizes at 18 bar for: (**a**) channel one, (**b**) channel two, and (**c**) channel three.

**Figure 13 sensors-23-09087-f013:**
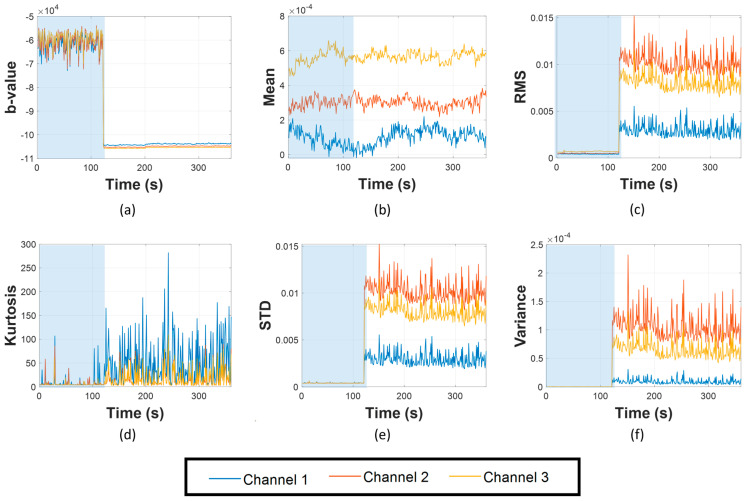
Comparison between AIIC and traditional indexes, with the blue region indicating normal working state, while the white one indicating leakage: (**a**) AIIC, (**b**) mean, (**c**) RMS, (**d**) kurtosis, (**e**) STD, and (**f**) variance.

**Figure 14 sensors-23-09087-f014:**
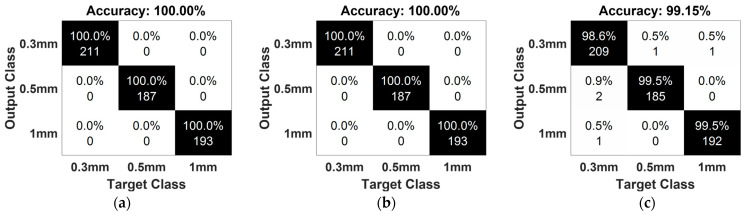
Result of leak size classification with different pressure levels: (**a**) 7 bar, (**b**) 13 bar, and (**c**) 18 bar.

**Table 1 sensors-23-09087-t001:** Hyperparameters for RF.

**Hyperparameters**	n_estimator	max_depth	min_sample_leaf	min_sample_split	criterion
**Value**	100	none	1	2	entropy

**Table 2 sensors-23-09087-t002:** Experimental setup description.

No.	Parameter	Description
1	Pipeline material	304 stainless steels
2	Pipeline thickness	6.02 mm
3	Pipeline outer diameter	114.3 mm
4	Sensor type	R15I-AST
5	Locations of sensor 1, 2, 3	0/1600/2500 mm
6	Leak location	800 mm
7	Total acquisition time	360 s
8	Normal state time/Leaking state time	120 s/240 s
9	Pipeline material	304 stainless steels

**Table 3 sensors-23-09087-t003:** Data description.

Test Number	Pressure Level	Working Conditions	Acquisition Time (s)	Number of Data Streams
1	7 bar	Normal/0.3 mm leak	120/240	3
2	7 bar	Normal/0.5 mm leak	120/240	3
3	7 bar	Normal/1 mm leak	120/240	3
4	13 bar	Normal/0.3 mm leak	120/240	3
5	13 bar	Normal/0.5 mm leak	120/240	3
6	13 bar	Normal/1 mm leak	120/240	3
7	18 bar	Normal/0.3 mm leak	120/240	3
8	18 bar	Normal/0.5 mm leak	120/240	3
9	18 bar	Normal/1 mm leak	120/240	3

**Table 4 sensors-23-09087-t004:** A comparison of leak size identification using the three methods under the pressure of 7 bar.

Method	Accuracy (%)			Average Accuracy (%)
	0.3 mm	0.5 mm	1 mm	
TD-CNN	86.9	67.0	96.6	88.5
Banjara et al. adaptation	78.8	74.7	77.5	77.0
The proposed method	100.0	100.0	100.0	100.0

**Table 5 sensors-23-09087-t005:** A comparison of leak size identification using the three methods under the pressure of 13 bar.

Method	Accuracy (%)			Average Accuracy (%)
	0.3 mm	0.5 mm	1 mm	
TD-CNN	68.5	75	100.0	79.6
Banjara et al. adaptation	81.6	95.9	100.0	92.2
The proposed method	100.0	100.0	100.0	100.0

**Table 6 sensors-23-09087-t006:** A comparison of leak size identification using the three methods under the pressure of 18 bar.

Method	Accuracy (%)			Average Accuracy (%)
	0.3 mm	0.5 mm	1 mm	
TD-CNN	64.4	67.6	100.0	75.9
Banjara et al. adaptation	97.0	45.0	100.0	87.0
The proposed method	98.6	99.5	99.5	99.2

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
