# Peer review of "Leak State Detection and Size Identification for Fluid Pipelines with a Novel Acoustic Emission Intensity Index and Random Forest"

_sensors, 2023, doi:10.3390/s23229087_

Round 1
Reviewer 1 Report
Comments and Suggestions for Authors
It is significant to investigate an effective method of pipeline leak detection leaking degree identification using AE signal. The English writing of this work is very good. It is acceptalbe after minor revision, there are some suggests for reference.
1. In equation(1) and equation(2), There are two "a", are they the same meaning?How to identify the value of variable "a"?
2. In the line 207, The BEAST is mentioned, does it mean a time-series signal includes Abrupt, Seasonal change and Trend? Which one represents Changepoint, Abrupt? But in equation (3), the part of Abrupt is not found. Can you explain the relationship between BEAST and equation(3)?
3. In section, the author mentions Random Forest, can you complement how to get prior probability when using BMA.
4. The sentence of the line 351-353 is difficult to understand.
5. The titles of Table4-6 are the same, pls check and modify them.
Author Response
"Please see the attachment.

Reviewer 2 Report
Comments and Suggestions for Authors
1. The introduction part is too long, please just focus on the leaking state monitoring problem and the advantage of AE monitoring for the leaking state.
2.“AE being a low-energy seismic activity” Usually, people regard AE as high-frequency seismic activity, although it has low-energy. Please check a more suitable description for the relationship between AE and seismic activity.
3.Please pay attention to the font specifications for labels in the formula; some of them require italics.
4.In Figure2, you should indicate what is occurrence.
5.Line 195, what is the “fluctuations”?
6.What is the purpose of using BEAST and the RF? On the other hand, you should give a clearer description about what is the input dataset, output dataset, and the workflow structure for this combination system (BEAST + RF). And before you use this method, you should at least give a figure to show what is the changepoint in your monitoring data, and an example of the b-value figure for the monitoring.
7.“amplitude threshold value at 10-2” What is the unit of 10-2? dB?
8.The number in Figure 6 is too small.
Reviewer 3 Report
Comments and Suggestions for Authors
This paper presents detection and size identification for fluid pipelines with a novel acoustic emission intensity index and random forest. Before considering it for publication, the following remarks should be addressed:
1- The abstract should be revised, including more details related to the technique used.
2- English typos and grammatical errors should be revised
3- Double check all equations, and some parameters were missed to be identified
4- The random forest parameters should be identified
5- Comparative study should be included to show the accuracy of presented technique
6- Some relevant work should be added to improve the introduction.
7- The conclusion should be improved, including the limitations of the presented technique.
Comments on the Quality of English Language
some parts should be improved.
Reviewer 4 Report
Comments and Suggestions for Authors
The present study discusses a framework for size identification and leak state detection in industrial fluid pipelines based on acoustic emission (AE) non-destructive testing. Specifically, the intensity index curve (AIIC) is adopted using the b-value filtered AE data. The probability of state transition from normal conditions to leakage is demanded by a Bayesian ensemble method. After detection, the method is able to provide a size identification of the leakage based on a classification problem solved by the aid of a machine learning random forest algorithm. The present study is quite well-written and organized, and the experimental results sound quite interesting. However, the authors must introduce the following revisions and changes to improve the scientific quality of the manuscript:
• Within the introduction, when nondestructive evaluation techniques are presented, and in order to improve and complete the state-of-art literature review on the use of non-destructive testing in engineering, the authors are suggested to take into consideration the present work as well:
Melchiorre, Jonathan, et al. "Acoustic emission and artificial intelligence procedure for crack source localization." Sensors 23.2 (2023): 693.
Chen, Zhuo, et al. "Detecting gas pipeline leaks in sandy soil with fiber-optic distributed acoustic sensing." Tunnelling and Underground Space Technology 141 (2023): 105367.
Rosso et al. "Comparative deep learning studies for indirect tunnel monitoring with and without Fourier pre-processing." Integrated Computer-Aided Engineering Preprint (2023): 1-20.
Li, Hao-Jie, et al. "Detecting pipeline leakage using active distributed temperature Sensing: Theoretical modeling and experimental verification." Tunnelling and Underground Space Technology 135 (2023): 105065.
• Figure 1, in the graphs on the right side of the flowchart please increase the font size to improve the readability of the image.
• Figures 4, 5, 6, 7, 10, 11, 12, 14, please substantially increase the font size to improve the readability of the graphs.
• Please check the entire manuscript to avoid typos and misspells.
• Table 1 reports optimal hyperparameters for RF, however, the authors do not explain how they obtained them. Some methods have been proposed in literature such as cross-validation, grid search, trial-and-error, etc. However, as reported in "Intelligent automatic operational modal analysis." Mechanical Systems and Signal Processing 201 (2023): 110669. The authors showed that due to the majority voting approach, the RF appears still stable in classification since it is less prone to overfitting phenomena. Therefore, the authors should revise the text part when describing Table 1 also commenting about the above-mentioned aspect and clarifying their optimal parameter tuning procedure.
• Despite the sentence “A dataset description for training/testing is presented in Section 3.2”, the Training and test sets are not explicitly described. Please revise this fundamental aspect. Did you use any cross-validation procedure as well?
• Table 4 and Table 5 make it hard to believe that the authors reached the perfect model since they claim that the accuracy of the proposed method is 100.00% (with double significant decimal figures!). Furthermore, they do not report clear information regarding the training and test set splitting. The test set has to be used as a blind validation of the model since the machine learning is fed with new previously unseen data. Please provide some convincing clarification to these suspicious results.
Round 2
Reviewer 2 Report
Comments and Suggestions for Authors
Most of the questions have been addressed, but I have one fundamental concern regarding the use of the b-value. This concern arises from the fact that whether it's earthquakes or rock fractures, there is indeed a logarithmic linear relationship in the occurrence frequency of different magnitudes. This is because the processes of earthquake formation and fracture formation are progressive, gradually moving from smaller to larger events. However, when it comes to leakage issues, the generation of a leakage source may not necessarily be a progressive process but rather a sudden event. Therefore, it is essential to be cautious when attempting to fit a reasonable linear relationship and obtain the b-value, as it may not always apply to the context of leakages due to the potential discontinuity in their source generation process.
Reviewer 3 Report
Comments and Suggestions for Authors
Can be accepted in the present form.
Reviewer 4 Report
Comments and Suggestions for Authors
The authors have provided all the explanations and the necessary revisions which were requested in the previous review step, thus improving the quality of the manuscript. Therefore, in summary, the results of the present paper are interesting, and I recommend the work be accepted for publication in the journal.